# Modification of a Cylindrical Mirror Analyzer for High Efficiency Photoelectron Spectroscopy on Ion Beams

**Francis Penent [1],\*** , **Denis Cubaynes [2,3]**, **Pascal Lablanquie [1]**, **Jérôme Palaudoux [1]**,
**Ségolène Guilbaud [2]**, **Olivier Moustier [2]**, **Jérôme Guigand [2]** and **Jean-Marc Bizau [2,3]**

[1] Laboratoire de Chimie Physique-Matière et Rayonnement (UMR 7614), Sorbonne Université, CNRS, F-75005 Paris, France; pascal.lablanquie@upmc.fr (P.L.); jerome.palaudoux@sorbonne-universite.fr (J.P.)

[2] Institut des Sciences Moléculaires d'Orsay (ISMO), CNRS, Univ. Paris Paris-Sud, Université Paris-Saclay, F-91405 Orsay, France; denis.cubaynes@universite-paris-saclay.fr (D.C.); segolene.guilbaud@universite-paris-saclay.fr (S.G.); olivier.moustier@universite-paris-saclay.fr (O.M.); jerome.Guigand@universite-paris-saclay.fr (J.G.); jean-marc.bizau@universite-paris-saclay.fr (J.-M.B.)

[3] Synchrotron SOLEIL, L'Orme des Merisiers, Saint-Aubin, BP 48, F-91192 Gif-sur-Yvette, France

\* Correspondence: francis.penent@upmc.fr

**Abstract:** An existing cylindrical mirror analyzer (CMA) that was initially equipped with eight channeltrons detectors has been modified to install large micro-channel plate detectors to perform parallel detection of electrons on an energy range corresponding to ~12% of the mean pass energy. This analyzer is dedicated to photoelectron spectroscopy of ions ionized by synchrotron radiation. The overall detection efficiency is increased by a factor of ~20 compared to the original analyzer. A proof of principle of the efficiency of the analyzer has been done for $Xe^{5+}$ and $Si^+$ ions and will allow photoelectron spectroscopy on many other ionic species.

**Keywords:** photoionization; electron spectroscopy; ion beam; synchrotron radiation

## 1. Introduction

Photoionization of atomic ions is a major importance to understand laser-produced laboratory and astrophysical plasmas (OPACITY and IRON projects [1,2]). Most experimental results on ionic species concern photoabsorption experiments for which photoion spectrometry has allowed the determination of absolute cross-sections for a variety of ions using the technique of merged ion and photon beams [3–5]. Recent insights have also been provided by other techniques, for instance by choosing $C^+$ ions and detecting $C^{4+}$ to unravel triple Auger decay [6], or by fluorescence spectroscopy with ion traps [7]—that is a nowadays an adaptation of the historical spectroscopy on plasma discharges—to study the emission spectra of ions. To go further in the interpretation of the ionization processes, it is necessary to understand how resonances, due for instance to the excitation of an inner-shell electron to a vacant orbital, will decay when different channels are possible (autoionization, Auger decay, cascade processes . . . ) to obtain partial cross sections. To unravel such processes, electron spectroscopy can be a very helpful technique. Here, contrary to the usual field of application of electron spectroscopy to neutral gas target or to surfaces, the difficulty comes from the very low ion density in an ion beam, typically six orders of magnitude lower than the density of a neutral gas target. It is hence very difficult to perform electron spectroscopy with ionic targets. While for ion spectrometry it is possible to use a long interaction region (~1 m) with overlapping ion and photon beams and to detect all the different charged states of the ion after magnetic deflection, it remains very difficult to improve the efficiency of electron spectroscopy with ion beams [8]. Due to reduced length of the source (~1 cm)

and acceptance solid angle (~1%) of the analyzer, the electron count rates are typically three or four orders of magnitude lower than the ion count rates. Moreover, the electron signal often appears on a high background resulting from ionization of the residual gas by the photon beam from collisions even in ultra-high vacuum conditions (~$10^{-10}$ hPa). Increasing the ion target density (if possible?) is not a real option since the charge of the ion beam will induce a shift and a broadening of the photoelectron peaks. Increasing the photon flux is possible by moving from synchrotron to X-ray free-electron laser (XFEL) sources but experiments remain nevertheless difficult [9]. Using $4\pi$ detection analyzers as magnetic-bottle electron spectrometers could appear as a possible option, but the Doppler-shift with ion velocity of the electron spectra becomes a real limitation [10] to perform electron spectroscopy with acceptable resolution. While it could be possible to use zero-degree electron spectroscopy [11–13] to limit the Doppler broadening, it seems that this possibility was never explored on synchrotron centers.

All these difficulties explain why experiments reporting electron spectroscopy with ion beams are scarce. For a long time, only two experimental results were published: on $Ca^+$ giant resonance [14,15] and on $Xe^+$ [16].

After the pioneering work on $Ca^+$ [14,15] at the Super-ACO synchrotron at Orsay, the cylindrical mirror analyzer (CMA) analyzer that was used to perform these experiments [17] was installed on the PLEIADES beamline of SOLEIL synchrotron to be used with the MAIA (multi analysis ion apparatus) set-up [18]. The analyzer equipped with eight channeltrons was used to study $Xe^{5+}$ ionization processes on 4d → nf resonances [19]. To extract the true electron signal from a high background, it was necessary to detect the product $Xe^{6+}$ ion in coincidence with the electron and to subtract the random coincidences. The method was efficient to study $Xe^{5+}$ resonances decay [19] with true electron count rates of about 1 e$^-$/s needing long acquisition time.

With the same set-up, it was further possible to extract the $Xe^+$(4d) photoelectron signal at 120.7 eV photon energy [20], the results were validated by cross-checking with core-valence double ionization of Xe [20] and have clearly shown that the first published results on $Xe^+$ [16] were not correct for some unknown reason. That proves, if necessary, how difficult electron spectroscopy with ion targets may be.

Although these recent results [19,20] are encouraging, with third-generation synchrotron sources, the electron count rates remain very small for most of the ions to study even when the photon energy is tuned on a resonance. Hence, long accumulation times are necessary that are not always compatible with the time allocated to a given experiment on the synchrotron (typically one or two weeks continuously).

## 2. Experimental Approach

This was the motivation to modify our CMA analyzer to increase its detection efficiency. This analyzer was first described in reference [15,17] and was designed to perform angle resolved photoelectron spectroscopy on ions using eight channeltrons. The source volume of the analyzer, defined by two ~6 mm slits in the two inner cylinders of the analyzer ($R_0$ = 39 mm and $R_{int}$ = 73 mm), is about 20 mm long (Figure 2 in [17]), although the main contribution is due to the central 6 mm zone which is not cut by the first slit. This analyzer can be inserted on the MAIA set-up, which is a permanent installation on the PLEIADES beamline at SOLEIL for the study of ionic species [18], and comes in place of the chamber used for ion spectrometry with merged ion and photon beams. The analyzer exit slit, in front of the channeltrons, was also 6 mm (see Figure 10 in [18]). The 8 channeltrons (~15 mm entrance cone) were positioned between the two inner cylinders of the CMA on the image focal ring (point to ring focusing [17]). To obtain a complete energy spectra on a broad energy range it is necessary to scan the analyzer voltage (difference between the potential of inner and outer cylinders).

Our simple idea was to open wide (>80 mm) the exit slit of the analyzer in order to replace the channeltrons by micro-channel plate detectors as it is usually done for the most frequently used 180° hemispherical deflector electron energy analyzer. The electron energy is deduced from the impact position of the electrons on the micro-channel plates (MCP) detector, thanks to position encoding systems (delay-line or phosphor screen with charge-coupled device (CCD) camera detector). With a cylindrical mirror analyzer, the focusing conditions (between the source and the image points), that

are a general rule for any electron analyzer [21], were defined in [17] to build this analyzer for a point to ring focusing, but they cannot be fulfilled for a point to cylinder focusing on a broad energy range. Although some prospective studies exist [22], where a point to cylinder focusing seems possible on a broad energy range for a modified CMA analyzer, this was not verified here. Nevertheless, since ultimate energy resolution is not the goal of our analyzer that aims to have the highest possible luminosity, we have relied on a simple electron trajectories simulation using SIMION® [23] to verify how the analyzer works in a given energy range.

### 2.1. SIMION Simulations with Cylindrical Symmetry

The real analyzer uses 15 guard rings that are polarized through a resistive chain to follow the $1/r$ variation of the potential. For sake of simplicity, an "ideal" CMA analyzer was simulated with SIMION in cylindrical symmetry with 1 mm SIMION graphic unit (gu). Due to the averaging property of the Laplace equation, no noticeable differences exist between the ideal and the real CMA at a few cm from the guard rings inside the analyzer, and the electron trajectories are absolutely similar. This "ideal" cylindrical analyzer allows to see how the electrons travel from the source point to the detector. We fixed the analyzer voltage to 0 V for the inner cylinder and $-24$ V for the outer cylinder, any other value is obtained by scaling all voltages. The potential between the two cylinders is $V(r) = (V_{in} - V_{out})\left(\frac{R_{out}R_{in}}{R_{out}-R_{in}}\right)\left(\frac{1}{r} - \frac{1}{R_{in}}\right) + V_{in}$ with ($R_{in}$ = 73 mm, $R_{out}$ = 226 mm).

For electrons created at the source point on the axis of the analyzer, we can see in Figure 1a that the focusing that is obtained for the central energy (29 eV) where the exit slit for the channeltrons detectors was initially positioned is no longer valid for all energies. The point to ring focusing conditions are not on a cylinder but rather on a cone frustum (best focus line). When the real size of the interaction volume (L~20 mm, R~1 mm) and the angular acceptance of the entrance slits are taken into account, the energy dispersion and the extension of the image for each energy remain however very good (Figure 1b). Since the radius of the ion and photon beams are about 1 mm, we do not consider the trajectories of electrons starting off axis that would spiral inside the analyzer. The analyzer is big enough ($R_{in}$ = 73 mm, $R_{out}$ = 226 mm, L = 500 mm) to make this problem not critical. However, it is necessary to keep the photon and ion beams on the analyzer axis to avoid a tilt of the image circle that would give different z on the detector for the same energy but different azimuth angle.

### 2.2. Mechanical Construction of the Analyzer

Since, at the moment, no cylindrical detector exist that could be installed after a cylindrical mesh ($R_{in}$, $V_{in}$) at the exit of analyzer, the cylindrical symmetry is broken when a detector is installed. Note that such detector could eventually have been developed with microsphere molding between two cylinders [24].

A first idea could have been to replace the eight channeltrons by eight MCP chevron assemblies located at ($0° + n \times 45°$; n = 0–7) angles with respect to the photon polarization (for direct determination of β asymmetry parameter at 0° and 90°). However, a cheaper option with only 6 MCP (L ~ 80 mm, w ~ 30 mm) was chosen ($0° + n \times 60°$; n = 0–5). The space available between the two inner cylinders allowed the insertion of 6 MCP assemblies on a hexagonal holder (see Figure 2). Compared to 8 channeltrons with 6mm slits the detection area is multiplied by (80/6) × (30/15) × (6/8) = 20, but the open area ratio (OAR) of the MCP being 70% a gain of ~14 is expected. This means than an increase in efficiency by about one order of magnitude and, consequently, experiments that would have need one week of continuous accumulation can be done in less than one day.

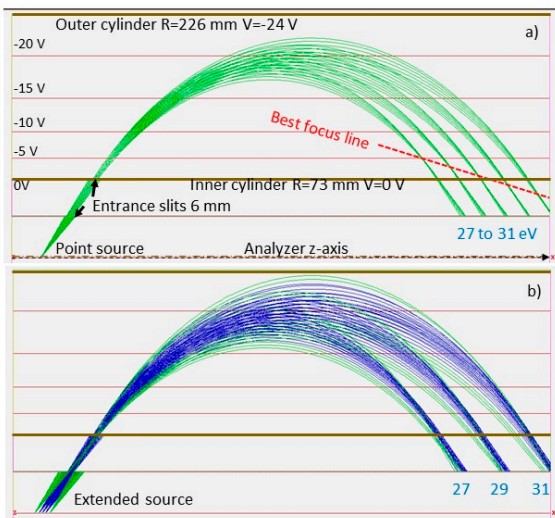

**Figure 1.** (**a**) SIMION$^®$ simulation of electron trajectories, starting from a source point, inside an ideal cylindrical analyzer for energies from 27 to 31 eV with $V_{in}$–$V_{out}$ = 24 V (the field varies as 1/r) and elevation angles from 50° to 60° (around the 54.4° magic angle). The two 6 mm entrance slits limit the acceptance angle. The focus on the inner cylinder is for K.E. ~ 29 eV. (**b**) Simulation for an extended source. The trajectories of electrons starting from the central 6 mm zone (blue trajectories) are not cut by the first slit, while electron from the two zones of 7 mm before and after are limited by the most inner slit. They induce an asymmetric broadening of the electron peak.

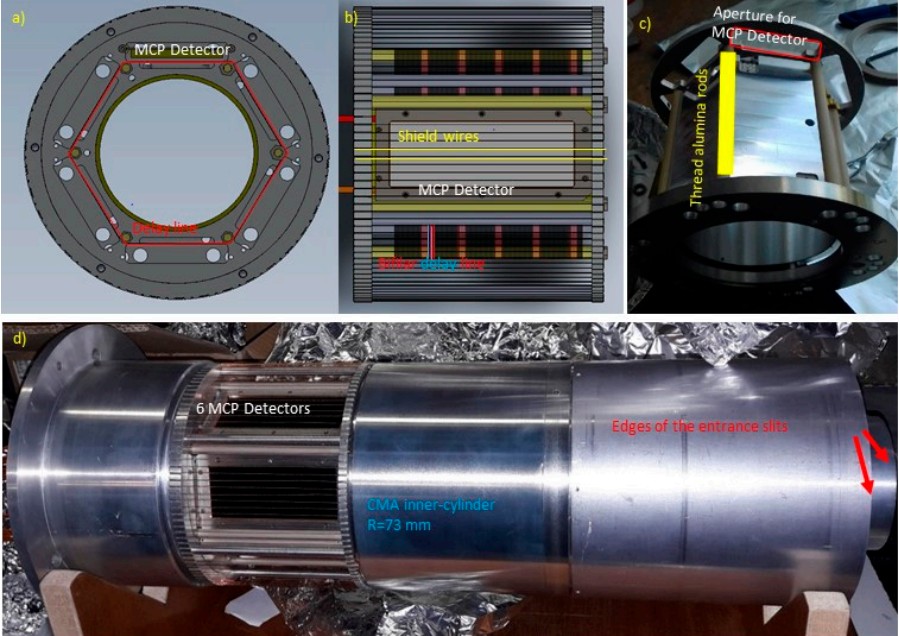

**Figure 2.** (**a**,**b**) The drawing of the detector. (**c**) hexagonal holder for the 0.5 mm threaded alumina rod on which the bifilar delay line is coiled. (**d**) The cylindrical mirror analyzer (CMA) inner cylinder with detectors installed. Longitudinal wires at the same potential allow the electrons to reach the micro-channel plates (MCP). The whole system is inserted in the CMA analyzer. Electrical connections (MCP polarization, delay line) go out on the left side of the image.

The position encoding along z is done with a bifilar delay line coiled around 6 alumina rods (Figure 2c) with 0.5 mm thread. Two copper wires (0.13 mm) were wound parallel simultaneously with a 1 mm step around the six alumina rods with a constant tension of 30 g along 85 mm to create a bifilar delay-line (see Figure 2a,b). A single turn corresponds to ~30 cm of wire length (1 mm along z and

~1 ns in time propagation), the total length of the line is ~25 m and corresponds to a time propagation of ~83 ns. The duralumin holder (Figure 2c) prevents flexion of the alumina rods when 85 turns are wound and avoids loosening of the coil. The electrical resistance of the wire is 30 ohms and does not induce a strong attenuation of the electron signal that propagates towards the two ends of the bifilar delay line. Position encoding was performed thanks to a time-to-digital converter (TDC-V4) [25] with 125 ps time resolution by measuring the time difference $(T_1 - T_2)$ of the signal that have propagated toward the two ends of the line. To be sure that the signals at the two ends correspond to the same electron the time $(T_1 + T_2 - 2T_0)$ must correspond to the delay line propagation time of 83 ns $\pm \Delta T$ (with $\Delta T \approx 2$ ns). $T_0$ corresponds to the time of arrival of the electron on the MCP and is given by the signals $S_{1-6}$ on the back of any of the 6 MCPs ($S_1$ OR $S_2 \ldots$ OR $S_6$) that initiates the TDC acquisition process. The signal from each channel plate chevron assembly was taken on the backside and decoupled from the high voltage by a capacitance out of the vacuum chamber.

As seen in Figure 1a, with 80 mm long detector it is possible to detect simultaneously energies from 27 to 31 eV. That corresponds to $\Delta E/E \approx 14\%$. The energy dispersion is linear with z in a very good approximation on the central 70 mm long zone of the MCP, however, due to some edge effects with the real detector (MCP holder thickness, delay-line termination), a ~5 mm zone on the two sides of the detector was securely excluded from the analysis by software filtering and limits $\Delta E/E$ to about 12%.

The rectangular MCPs were manufacture by Hamamatsu® and are $87 \times 37$ mm outer dimensions with $81 \times 32.5$ mm active area. The detector consists of two MCPs in chevron position that are inserted in a PEEK holder and polarized thanks to thin 0.1 mm rectangular copper frames. Metallic frames of $98 \times 48$ mm (visible in Figure 2b,d) allow to secure the two MCPs, and the insertion of the detector between the two inner cylinders with a groove in PEEK rods. Although the hexagonal shape of the detector does not match the cylindrical symmetry, the distance (~1 cm) between the shield wires (Figure 2b), that are at the potential of the inner cylinder of the CMA, and the front MCP, on which the electron are accelerated to ~200 eV, is not critical and a circle is projected as a straight line on the MCP. Six detectors can be installed at 60° from each other, but, unfortunately, due to a crack in one set of MCPs only 5 could be installed. The whole system (Figure 2d) with the detectors is than inserted in the CMA analyzer.

The insertion of the detector into the analyzer can be seen in Figure 3.

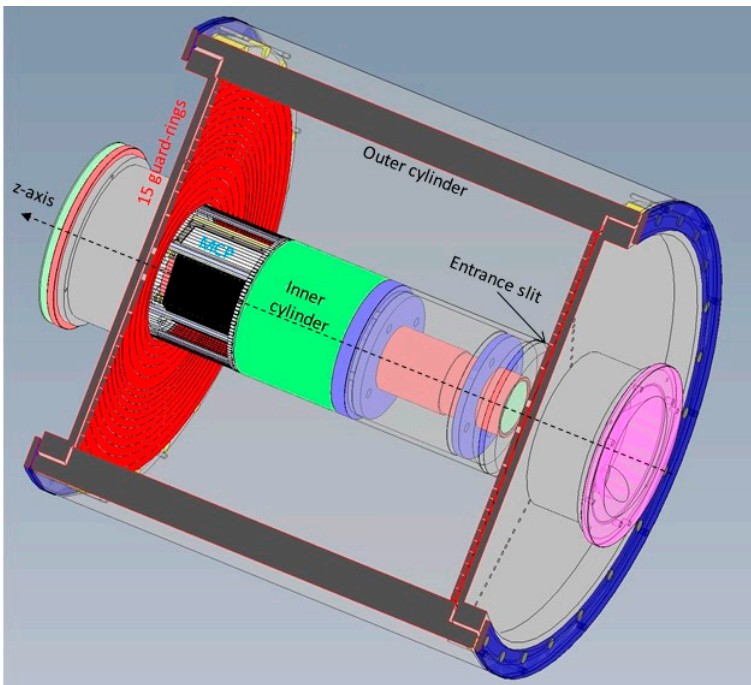

**Figure 3.** 3D sectional plan of the real CMA analyzer showing all the electrodes and the insertion of the MCP detectors. The ion beam travels along the z-axis inside the most inner cylinder $R_0$.

### 2.3. Optimization of the Analyzer

Since a single delay-line is used to detect the signal from different MCPs, the potential of the back MCP (about 1800 V) must be the same for all MCPs in order to optimize the detection of the electron cloud on the wires that are polarized at +200 V and +150 V with respect to the MCP back. The MCP must have similar impedances and signal amplification factor. They were manufactured in the same process and the impedances were measured to match the same characteristics. A single detector was tested in a first step of the analyzer development and the MCP characteristics were slightly different from the next ones. Consequently, the amplification of the different MCP could be different when the same potential is applied to polarize the MCPs. This makes the adjustment of discriminator threshold critical.

Another difficulty is due to the length of connecting cables from the vacuum flange to the MCP (about 1 m) that are not exactly matched in impedance with the bifilar delay line. This induces some rebounds in the electron signal. Hopefully, since the electron count rates remain low it was possible to broaden the signal to 200 ns to kill the rebounds in the logic NIM or TTL signals. Another problem is some cross-talk between the different MCP detectors. In the present configuration, is would be rather difficult to deduce beta parameter from the count rate on different MCPs positioned at different azimuthal angles.

The signal at each end of the delay line is a differential signal because the electrons are collected on the higher potential wire (cf. Roentdek® [26]) and is very clean. The signal from the back MCP is noisier. The ultrahigh vacuum conditions and the limited space for the MCP assemblies do not allow to use in-vacuum capacitors and resistors close enough from the detector to filter out the MCP signal. The connection of signal cables is also critical because tin solder cannot be used due to baking of the analyzer at more than 100 °C to reach ultrahigh vacuum. This baking was also critical for the delay line detector and the copper wires were relaxed and eventually short-circuited after baking operation.

Nevertheless, most of these little (but time consuming due to ultra-high vacuum constraints) problems could be reduced and the analyzer operated as expected.

The calibration of the analyzer is done by introducing neon gas target in the chamber to a pressure of $10^{-7}$ hPa. By changing the analyzer voltage and/or the photon energy, the 2p photoelectron peak (Figure 4) is moved from one edge to the other of the MCP and allows a precise calibration of the electron energy with the position on the detector. This procedure allows also to verify that no major distortion due to bad focusing conditions occur. Since the channel plates were only detection quality ones, and not imaging quality for cost limitations, it is necessary to move the photoelectron peak on all positions on the MCP to have an average detection efficiency on the whole area. The position is linearized as a function of the analyzer voltage and an electron spectra is collected (Figure 4).

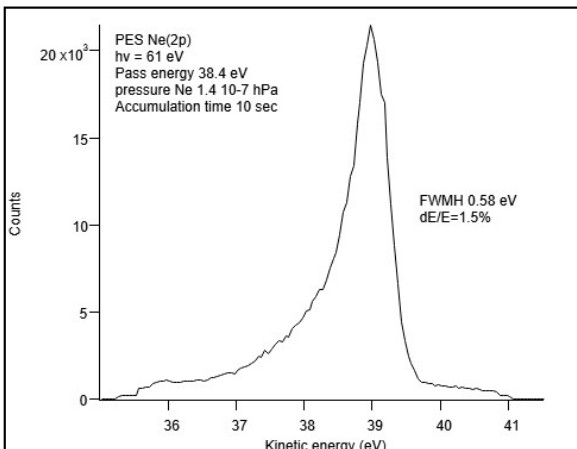

**Figure 4.** Ne 2p photoelectron peak. The resolution $\Delta E/E$ is 1.5% thanks to the 1 mm precision encoding with the delay line. The peak asymmetry towards lower energy can be explained by the focusing at lower z of electrons created out of the central zone seen in Figure 2b.

The control of all the experimental parameters (photon energy, analyzer voltages) and the acquisition is done with different Labview procedures [18].

After the ionization event, the time of arrival of the electrons to the detector and of the signal at the end of the delay line are less than 100 ns, but in order to extract the true electron signal the product ion is detected in coincidence with the photoelectron. The product ions are detected after magnetic q/m deflection and energy analysis by an electrostatic deflector before detection by MCPs with a typical efficiency of 50% (if the ions are fast enough). A phosphor screen and a CCD camera are used to center the product ions on the MCP by adjusting the current in the analyzing magnet and the deflector potentials. The time of flight (TOF) of ions to the detector depends on the velocity of the ion and is typically a few μs, much longer than electron TOF. The TDC window is opened for a time long enough for the arrival of the ion to the detector. In practice this time is also a limit for the electron count rate since no other electron should initiate another timing measurement.

## 3. Experimental Results

### 3.1. Results on $Xe^{5+}$

$Xe^{5+}$ 4d inner-shell photoionization was measured in a previous experiment [19] with the original configuration of the analyzer with 8 channeltrons. The experiment was repeated with the modified analyzer, after baking of the CMA, during another experimental session of 21 shifts. In Figure 5 we compare the two spectra on $Xe^{5+}$ ions [19] at 94.4 eV photon energy (4d → 4f resonance) with the 8 channeltrons detector (lower panel Figure 5b) and with the analyzer equipped with 5 MCPs detectors (upper panel Figure 5a). This last spectrum was obtained by scanning the CMA pass energy between 26 and 37.5 eV with 0.5 eV step. We observe two lines corresponding to the $Xe^{6+}$ $5s^2$ final state starting from $Xe^{5+}$ 5p $^2P_J$ J = 1/2 and 3/2 levels: $Xe^{5+}[Kr]5s^25p\ ^2P_{1/2;3/2} + h\nu(94.4\ eV) \rightarrow Xe^{5+}4d^95s^25p4f \rightarrow Xe^{6+}\ 5s^2 + e^-$.

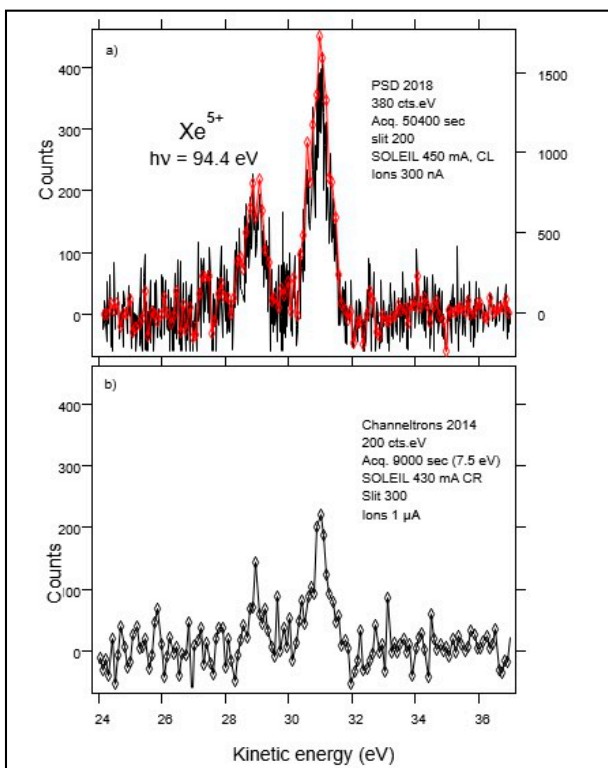

**Figure 5.** Comparison of electron spectra in $Xe^{5+}$ (4d → 4f) resonance with different detection systems: (**a**) Position sensitive detector with 5 MCPs versus (**b**) 8 channeltrons. The energy step is 20 meV in (**a**) and 100 meV in (**b**). To make the comparison easier, the red curve in (**a**) was summed on 5 points to compare the counts rate (right scale) with (**b**) with the same 100 meV steps.

It was obtained in 14 h with much better statistics and also better energy resolution because the resolution in position with the delay line encoder is 1 mm while the slit in front of the channeltron was 6 mm. The energy step coherent with the resolution in position in Figure 5 was 100 meV with the channeltron and 20 meV with the MCPs.

The quality of the spectrum is clearly improved, the gain in efficiency is close to one order of magnitude but remains difficult to precisely quantify because the overlap between the photon and ion beams was not measured in these experiments and many other parameters (ion current, acquisition time, photon flux, ... ) changed between the two experiments taken with 4 years interval.

### 3.2. Results on Si$^+$

In Figure 6 we present the electron spectra on the strongest $2p \rightarrow 3d$ resonance of Si$^+$ at 111.74 eV [27] corresponding to Si$^+$ $2p^{-1}3s^23p3d$ $^2D_{5/2}$ configuration. This spectra was first obtained with the analyzer equipped with only one MCP assembly. We can see on this spectra that the preferential decay channel correspond to an Auger decay with a spectator 3d electron leading to Si$^{2+}$ 3s3d $^3D$. This dominant decay channel corresponds to a filling of the 2p hole by a 3s electron with ejection of the 3p electron. There is no evidence of participator Auger decay of the 3d electron giving Si$^{2+}$ ion in its ground state $2p^63s^2$ $^1S$ that would involve filling of the 2p hole by a 3p electron and ejection of the 3d electron. It seems however that the 3s3p $^1P$ state can be reached by filling of the 2p hole by a 3s electron and ejection of the 3d electron. Some other peaks, at lower kinetic energy, correspond certainly to Auger decay with shake-up of the 3d electron.

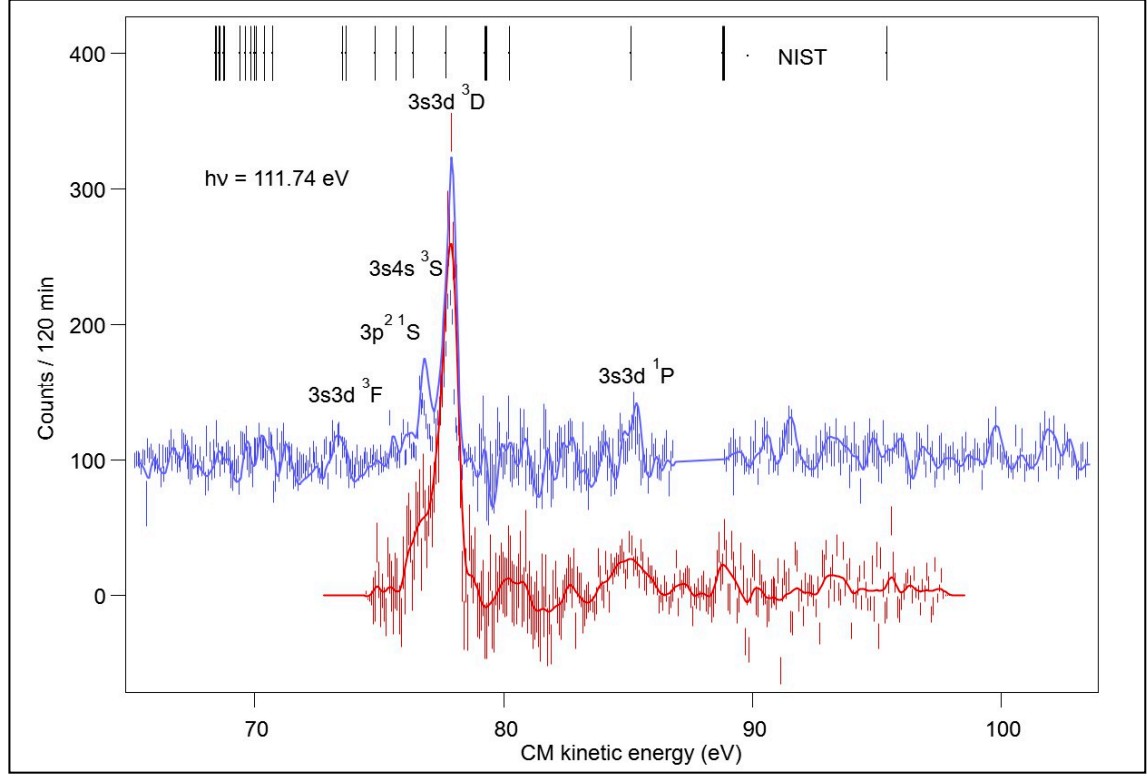

**Figure 6.** Electron spectra on Si$^+$ $2p \rightarrow 3d$ $^2D_{5/2}$ resonance (two different energy scans). The black lines correspond to the energy levels from NIST for Si$^{2+}$ [28].

This confirms that the electron spectra provide essential additional information to the results obtained by ion spectrometry [27]. The final Si$^{2+}$ states are excited states that can only decay by emission of a photon that could be detected by optical spectroscopy.

## 4. Conclusions

We have proved by the present results on a few examples that the concept of parallel energy detection with a CMA analyzer is a very good option although the optimal focusing is not achieved for all electron energies. We have modified our CMA analyzer to install large MCPs detectors. The $Si^+$ and $Xe^{5+}$ examples show that electron spectroscopy can be performed in a reasonable time when absolute cross section are of tens of Mb and not only giant resonances. This could be extended to molecular ions of astrophysical interest as $CH^+$, $SiH^+$, $OH^+$ [29] and possibly to $CH_n^+$, $SiH_n^+$. Although some improvements are still possible and desirable, this analyzer is a good tool to perform photoelectron spectroscopy with ion beams with an angular acceptance about 0.7% of $4\pi$ solid angle around the 54.4° magic angle. One practical problem is the need of impedance matching between all MCP assemblies to insure the same detection efficiency in order to determine partial angular (azimuthal) distributions of photoelectrons. Hence the global cost of the detection system may also be an issue. Some improvements remain possible to precisely adjust the detection efficiency of the 6 MCPs set-up. By installing a grid just in front of the MCP at a potential of ~200 V, it is possible to adjust independently the voltage of the front MCP (typically 200 ± 50 V) to reach the same amplification for all the detectors, since it is necessary to keep the same voltage on the back of all the MPC to have the same extension of the electron cloud on the delay line. The decoupling of the signal—to avoid cross-talk that is also a crucial issue to determine angular distribution—would need vacuum capacitors and resistors compatible with an ultra-high vacuum and some miniature high voltage connectors (not commercially available) providing reliable connections to avoid any soldering. Another difficulty with ultrahigh vacuum is the need of baking that can loosen fixations. When one week is allocated for an experiment it is not possible to make any repair in a short delay with these constraints. To obtain the precise angular distribution, it is certainly possible to use a ceramic printed circuit board to create a bifilar delay line in a direction perpendicular to the wires. There is enough space available between the wires and the hexagonal holder to insert such a system. This could be an ultimate development of the position sensitive detector for z and θ determination.

**Author Contributions:** Conceptualization: F.P., D.C. and J.-M.B., software: O.M. and J.-M.B.; formal analysis: J.-M.B., P.L., S.G. and D.C.; investigation: F.P., D.C., P.L., J.P., S.G., O.M., J.G. and J.-M.B.; resources: J.G. and O.M.; data curation: J.-M.B., D.C. and F.P.; writing—original draft preparation, F.P.; writing—review and editing, F.P., D.C. and J.-M.B.; visualization, F.P., D.C., J.G. and J.-M.B., supervision, F.P., D.C. and J.-M.B.; project administration, F.P., D.C. and J.-M.B.; funding acquisition, F.P., D.C. and J.-M.B. All authors have read and agreed to the published version of the manuscript.

**Funding:** This research has been done within the LABEX Plas@par project, and received financial state aid managed by the Agence Nationale de la Recherche, as part of the programme "Investissements d'Avenir" under the reference ANR-11-IDEX-0004-02.

**Acknowledgments:** The experiment was performed at SOLEIL Synchrotron (France) at the PLEIADES beam line, with the approval of the SOLEIL Peer Review Committee (Project N° 20160251, 20170042, 20181434). We thank the PLEIADES beam staff for help during the experiments.

**Conflicts of Interest:** The authors declare no conflict of interest. The funders had no role in the design of the study; in the collection, analyses, or interpretation of data; in the writing of the manuscript, or in the decision to publish the results.

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
