# Peer review of "Modification of a Cylindrical Mirror Analyzer for High Efficiency Photoelectron Spectroscopy on Ion Beams"

_atoms, doi:10.3390/atoms8040063_

Round 1

Reviewer 1 Report

The authors describe the technical improvement of an electron spectrometer to be used in synchrotron-based photoionization studies with ion beams. As the authors discuss in some detail, such experiments are challenging because ion beams are very dilute targets which generally results in low electron count rates. In addition, the Doppler shift associated with the movement of the ions potentially deteriorates the electron energy resolution. The authors are among the pioneers of such tedious studies. Here, they describe several design changes to their electron analyzer that aim at higher electron count rates.  They present results of electron optical simulations and compare new experimental results directly with their earlier work. This comparison clearly shows that the experimental conditions could be significantly improved.

The paper is sound and the overall presentation is very clear. I only disagree with the statement that electron spectroscopy is "mandatory" for a detailed understanding of photoionization processes (line 32). I rather see electron spectroscopy as a potentially powerfull technique for gaining more insight. In practice, the technical challenges still remain considerable as is also demonstrated in the present manuscript. Besides, more insight into the photoionization process was also provided by other techniques as, for example, by a clever choice of ionic species as shown by Muller et al in their study of the triple Auger process (PRL 114, 013002 (2015)). Another technique that allows further insight is fluorescence spectroscopy (e.g. Kuhn et al, PRL 124, 225001 (2020)) which, at present, seems to be more developed than electron spectroscopy of ions. The authors should modify their introduction and formulate more carefully. The above mentioned papers should be cited to put the present work into a proper scientific context.

Another remark concerns Ref 4. This review has been very recently superseded (Schippers and Muller, Atoms 8, 45 (2020)). It would be appropriate to rather cite the more recent paper and not the earlier one.

I recommend that the present manuscript be published after the authors having made the minor changes outlined above.

Author Response

We thank the referee for the careful reading of the paper and for the constructive suggestions.

Following the comments of referee n°1 we have slightly modified the introduction and added the suggested references.

Reviewer 2 Report

This is a very interesting manuscript in which the authors share extensive knowledge in the field of laboratory atomic physics.  The authors describe the modification of a cylindrical mirror analyzer for high efficiency photoelectron spectroscopy on ion beams. This work is very useful for the scientific community and surely deserves publication in Atoms but with few requests.

Add sentence in the Sec. Conclusion describing what is achieved with these improvements and why it is important for the application in laboratory and astrophysical plasmas or similar. Highlight what you have achieved. Also, some self-criticism should be added. This should be added also in Introduction.

Page 2: cylindrical mirror analyzer (CMA) abbreviation  should be defined also in the main text i.e. line 52 "... Orsay, the CMA ..." => "... Orsay, the cylindrical mirror analyzer (CMA) ..."

-line 43, also for  XFEL => X-ray free-electron laser (XFEL)

-line 87, define abbreviation MCP

Maybe is good idea to add The list of abbreviations at the end of the text.

Fig. 4 quality is poor. The same goes for Fig 5 and Fig 6. Also, you have two Fig 5! Please correct this (line 261)  

The Author Contributions, Conflicts of Interest are missing.

If you use in References ISO 4 abbreviated journal name, then make it uniform and correct in

Ref 1  AIP Conference Proceedings = > AIP Conf. Proc.

Ref 2   A&A => Astron. Astrophys.

Author Response

We thank the referee for the careful reading of the paper and for the constructive suggestions.

We have also followed most recommendations of referee n° 2. (abbreviations, conclusions, contribution, …). We believe that self-criticism is present in this paper, since we have mentioned what should be improved in the present development of our analyzer? Electron spectroscopy is a complementary tool of ion spectrometry and of fluorescence studies, to study photoionization of ions. We could have mentioned a recent reference Mercadier et al, PRL 123, 023201 (2019) https://doi.org/10.1103/PhysRevLett.123.023201 where some authors of the present paper have shown, by Auger electron spectroscopy, that an inversion of population between Xe2+ sates was responsible for superfluorescence in the EUV region. This population inversion in highly charged ions was also at the basis of X-ray laser following plasma creation by high power lasers. Electron spectroscopy on ion beams could also reveal clearly which Auger decay can contribute to such population inversion. Since our paper does not pretend to be an extensive review and aims to show what will be possible concerning electron spectroscopy with ion beams, we believe that enough details are given that would allow further developments.

We hope the figures are clear enough (they were fuzzy in the first submission but have been modified since).

Reviewer 3 Report

The authors are reporting on the modification of a CMA spectrometer as to include MCP detection (imaging the spectrum) instead of channeltron (scanning the spectrum). This act resulted in an improvement of the detection efficiency by more than an order of magnitude. I admire their courage and patience towards this change while obviously fighting for funding it and I feel sorry for the loss of one of the MCPs.  The paper although quite technical has new results and therefore is suitable for publication in ATOMS. There are some questions and comments that should bean answered first which can be found below.

Authors claim that their SIMION study is “for the sake of simplicity”. I do not agree with this opinion. SIMION under proper geometry building can show details otherwise too difficult to envisage (see for example a cure for fringing fields in a similar hemispherical spectrograph in Nucl. Instum. Methods Phys. Res. A 440, 462 (2000). As I state in my next comment the authors would benefit form a detailed simulation of their spectrograph in SIMION. For know I think they should include in their manuscript a few details about their SIMION study, as for example the ratio of gu/mm and the total size of the volume.

Figure 5. I think that in the discussion about the comparison of the spectra in this figure the authors do not stress enough the fact that their new spectrum even though it seems to be in the same scale in the  y axis (counts), it has many more counts. The old data have a step of 0.5 eV while the new data perhaps a step of 0.05 eV. I suggest that they do an appropriate energy summing of the new spectra so that the step is the same as the old one and put it in the same top graph. Then the result of the enhancement of the detection efficiency would be clearly visible.   

I do not seem to understand the polarizing of the MCP to 200 V. It is known that the electron detection efficiency is maximum around 300 eV and therefore MCP voltage schemes should fulfill this. I guess this is the polarizing of the MCP to 200 V. But then my question is how this high value voltage for the low energy electrons detected affect the operation of the spectrograph with respect to energy shifts, fringing fields, etc. I think that a detailed SIMION3D study would benefit the investigation of the operation conditions of the spectrograph and its future operation.   

Additional bibliography:

Line 47. A standard detailed paper for zero-degree electron spectroscopy is the following:

Zouros, T.J.M.; Lee, D.H. Zero Degree Auger Electron Spectroscopy of Projectile Ions. In Accelerator-Based Atomic Physics Techniques and Applications; Shafroth, S.M., Austin, J.C., Eds.; American Institute of Physics: Woodbury, NY, USA, 1997; Chapter 13, pp. 426–479.

Line 84-86. From the historical point of view the authors should give credit to similar trials to incorporate MCPs in standard spectrometers as the hemispherical detector analyzer. One of the most detailed work on this can be found in J. Electr. Spectr. Rel. Phenom. 163, 28 (2008).  

General remarks:

Leave space after the reference in brackets.

Leave space between numbers and units.

Remove the parenthesis when referred to multiple figures e.g. “Fig.2a)”  should be “Fig. 2a”.

ref. -->  Ref.

Justify centered all figures

Minor corrections:

Figure 1 should have axes as in the text the authors refer to the z axis.

Line 14.  channeltrons --> chanelltron

Line 24.  do -->  to

Line 33, 38.  order --> orders

Line 50. only a two --> only two

Line 69.  Remove greek “m” from continuously

Line 72. reference --> Refs.

Line 97. Remove “For sake of simplicity “see my comments on SIMION.

Line 98, 114.  A better wording for “perfect” is “ideal” which is more common in the literature.

Line 117. B) --> b)

Line 173, 203. Do not change paragraph.

Line 245. Insert period at the end of the sentence.

Author Response

We thank the referee for the careful reading of the paper and for the constructive suggestions.

We have taken into account some remarks of referee n° 3. Some points have been clarified and details  corrected. We have specified what we mean by “for sake of simplicity” in our SIMION simulations. A more realistic simulation of our CMA with the guard rings was done (in cylindrical symmetry) but it gives exactly the same results for the electron focusing that an ‘ideal' CMA. It was not possible to simulate a 3D analyzer with enough spatial resolution because simulating wires (and field penetration) is a real issue and in SIMION an electrode of zero thickness is a perfect grid. The 3D simulation would be necessary to include the detector inside the analyzer, but since the experiment was successful, it should only confirm our measurements.

The situation for a CMA is different from the Hemispherical analyzer (HDA) where the extended PSD detector is positioned perpendicular to the equipotential lines. Here, the inner cylinder is equipotential, and to our knowledge (?) it is the first time a CMA is equipped with PSD. We have prefered not to give any reference to PSD with HDA because it is commercially available for a long time.

We have added the suggested reference for zero degree electron spectroscopy and also another one that is informative and was published earlier (if the historical point of view is considered) and also because the book article reference suggested by the referee could be difficult to find.

We think the 200V polarization of the front MCP in understandable (and was enough to detect efficiently the electrons).

We have modified Figure 5 to compare the counts with the same energy step. Since many parameter changed between the two experiment a precise quantitative comparison is tricky.